*Proc. R. Soc. B* **287**: 20200620.

microbiology, palaeontology, ecology

planktonic foraminifera, dinoflagellate symbiosis, ammonium assimilation, ammonium recycling photosymbiosis

**Author for correspondence:**
Charlotte LeKieffre
e-mail: charlotte.lekieffre@gmail.com

# Ammonium is the preferred source of nitrogen for planktonic foraminifer and their dinoflagellate symbionts

Charlotte LeKieffre[1,2,†], Howard J. Spero[3], Jennifer S. Fehrenbacher[4], Ann D. Russell[3], Haojia Ren[5], Emmanuelle Geslin[2] and Anders Meibom[1,6]

[1]Laboratory for Biological Geochemistry, School of Architecture, Civil and Environmental Engineering (ENAC), Ecole Polytechnique Fédérale de Lausanne (EPFL), Switzerland
[2]UMR CNRS 6112 - LPG-BIAF, Université d'Angers, 49045 Angers Cedex, France
[3]Department of Earth and Planetary Sciences, University of California Davis, Davis, CA 95616, USA
[4]College of Earth, Ocean, and Atmospheric Sciences, Oregon State University, Corvallis, OR 97331, USA
[5]Department of Geosciences, National Taiwan University, Taipei, Taiwan
[6]Center for Advanced Surface Analysis, Institute of Earth Sciences, University of Lausanne, Switzerland

(iD) CL, 0000-0002-9200-7925

The symbiotic planktonic foraminifera *Orbulina universa* inhabits open ocean oligotrophic ecosystems where dissolved nutrients are scarce and often limit biological productivity. It has previously been proposed that *O. universa* meets its nitrogen (N) requirements by preying on zooplankton, and that its symbiotic dinoflagellates recycle metabolic 'waste ammonium' for their N pool. However, these conclusions were derived from bulk $^{15}$N-enrichment experiments and model calculations, and our understanding of N assimilation and exchange between the foraminifer host cell and its symbiotic dinoflagellates remains poorly constrained. Here, we present data from pulse-chase experiments with $^{13}$C-enriched inorganic carbon, $^{15}$N-nitrate, and $^{15}$N-ammonium, as well as a $^{13}$C- and $^{15}$N- enriched heterotrophic food source, followed by TEM (transmission electron microscopy) coupled to NanoSIMS (nanoscale secondary ion mass spectrometry) isotopic imaging to visualize and quantify C and N assimilation and translocation in the symbiotic system. High levels of $^{15}$N-labelling were observed in the dinoflagellates and in foraminiferal organelles and cytoplasm after incubation with $^{15}$N-ammonium, indicating efficient ammonium assimilation. Only weak $^{15}$N-assimilation was observed after incubation with $^{15}$N-nitrate. Feeding foraminifers with $^{13}$C- and $^{15}$N-labelled food resulted in dinoflagellates that were labelled with $^{15}$N, thereby confirming the transfer of $^{15}$N-compounds from the digestive vacuoles of the foraminifer to the symbiotic dinoflagellates, likely through recycling of ammonium. These observations are important for N isotope-based palaeoceanographic reconstructions, as they show that $\delta^{15}$N values recorded in the organic matrix in symbiotic species likely reflect ammonium recycling rather than alternative N sources, such as nitrates.

[†]Present address: Cell and Plant Physiology Laboratory, University of Grenoble Alpes, CNRS, INRAE, CEA, Grenoble, France.

## 1. Introduction

Planktonic foraminifera are important contributors to primary production in open ocean oligotrophic environments because they are often associated with large populations of photosynthesizing symbiotic algae such as dinoflagellates [1,2]. It was shown that assemblages of planktonic foraminifera and acantharia contribute an average of 5% to annual primary production in the Sargasso Sea [1]. The species *Orbulina universa* alone is estimated to contribute to about 1% of the total inorganic carbon fixed by all primary producers, a value that could be much higher in patches of high foraminiferal density [2]. Microsensor experiments also showed that carbon assimilation by photosynthetic symbionts of this species exceeds the

carbon requirements for the foraminifera host cell and symbiont growth [3]. In oligotrophic regions, dissolved inorganic nitrogen (e.g. $NH_4^+$ or $NO_3^{2-}$) is scarce and often limits biological productivity [4]. Bulk $\delta^{15}N$ data from cultured foraminifera, measured after $^{15}N$-spiked experiments, showed N-compound translocation from the endosymbionts to the foraminifera host cell [5]. When nitrate is not limiting in the environment, model calculations revealed that up to 57% of the foraminiferal N could be translocated from the symbionts, and even up to 90–100% for model calculations taking into account nitrogen uptake by the symbionts from the recycled ammonium pool. However, several studies have concluded that symbiont-bearing foraminifera must meet their nitrogen requirements via capture and assimilation of zooplankton prey N, because the concentrations and diffusion rates of dissolved ammonium or nitrate are insufficient to explain their growth rate. In this regard, $\delta^{13}C$ values measured on different amino acids from the symbiotic species *Orbulina universa* are consistent with two isotopically distinct sources of carbon and nitrogen, isotopically heavy metabolic carbon and nitrogen from its symbionts and relatively lighter carbon and nitrogen from the diet [6,7]. It is also likely that planktonic foraminifer can assimilate inorganic nitrogen from seawater if it is available [2,5,8] because ammonium assimilation has already been reported in a benthic species with inactive kleptoplasts [9].

Determining the source(s) of nitrogen for foraminifera associations is particularly important when interpreting planktonic foraminifera $^{15}N/^{14}N$ data from fossil shells [10]. In these shells, foraminifera-bound $\delta^{15}N$ (FB-$\delta^{15}N$) was measured by extracting organic matrix protein N that is tightly sequestered within the foraminiferal calcite chamber walls. Ren *et al.* [11,12] demonstrated that the dinoflagellate bearing foraminifers, *O. universa*, *Trilobus sacculifer*, and *Globigerinoides ruber* record similar FB-$\delta^{15}N$ values in modern ocean and recent sediments from the past 40 ka, and their FB-$\delta^{15}N$ values are similar to the $\delta^{15}N$ of nitrate sources. By contrast, in the co-occurring species *Globigerinella siphonifera* (=*G. aequilateralis*), which hosts small coccoid pelagophyte symbionts [13], FB-$\delta^{15}N$ values were systematically 1–2‰ higher compared with the dinoflagellate-bearing species even though the four species appear to feed on similar zooplankton prey [14]. A clearer understanding of the nitrogen uptake and protein synthesis pathways will help determine whether such offsets represent trophic effects or species differences in symbiont-host physiology.

Inorganic carbon assimilation by the symbiotic dinoflagellates of *O. universa* has been quantified in laboratory experiments using microelectrode techniques [3,8]. More recently, LeKieffre *et al.* [15] used a combination of transmission electron microscopy (TEM) and nanoscale secondary ion mass spectrometry (NanoSIMS) quantitative imaging to document inorganic carbon flow from symbiont photosynthetic assimilation through lipid translocation to the foraminifera cell in a time-series experiment.

Here, we present results from a second TEM–NanoSIMS time-series experiment using seawater with $^{15}N$-enriched $NH_4$, $NO_3$, and $^{13}C$ and $^{15}N$-enriched food (*Artemia*, brine shrimp) to explore the different nitrogen assimilation pathways in the *O. universa*-symbiont system. The combination of TEM and NanoSIMS was used in combination with stable isotope incubations to follow the assimilation of carbon and nitrogen from different sources within the foraminiferal cell and its symbiotic microalgae across a full day/night cycle. The main objective of this study was to visualize and quantify nitrogen assimilation and translocation

between dinoflagellate symbionts and *O. universa* host cell when provided with different nitrogen sources to better understand the nitrogen flow dynamics in symbiotic associations.

## 2. Material and methods

### (a) Sampling

All specimens of *Orbulina universa* used in the experiments were hand-collected by scuba divers from surface waters (2–10 m depth), approximately 1–2 km offshore Santa Catalina Island (California, USA) in the San Pedro basin on 25 August 2015. Individuals were collected in glass jars (120 ml) and transported within an hour to the University of Southern California, Wrigley Marine Science Center. In the laboratory, each individual was measured, checked for vitality (i.e. chambers filled with cytoplasm), and for the presence of symbiotic dinoflagellates and/or food particles in the spines (see electronic supplementary material, table S1), and then transferred to a new glass jar filled with filtered (0.45 µm) seawater. Both 'juvenile' (pre-sphere, trochospiral shell) and 'adult' (specimens with a final spherical chamber) specimens were used in order to obtain sufficient replicates for the experimental matrix (see electronic supplementary material, table S1). Except for the feeding experiment (Exp. 3), all foraminifera were fed an unlabelled one-day-old hatched *Artemia* nauplius at 16.00 on the day the specimens were collected to maximize their survival rate during the experiment phase. This includes specimens that were collected with food particles in their spines.

### (b) Isotopic enrichment of *Artemia salina*

$^{15}N$- and $^{13}C$-labelled *A. salina* used in Exp. 3 were prepared according to the protocol described by Krueger *et al.* [16] detailed in the electronic supplementary material, text S1.

### (c) Experimental design

Three isotopic labelling experiments were initiated the day after collection. In Exp. 1, specimens of *O. universa* were incubated for 18 h in seawater that was double-spiked with $^{15}NH_4^+$ and $H^{13}CO_3^-$. For Exp. 2, specimens of *O. universa* were incubated for 18 h in seawater that was double-spiked with $^{15}NO_3^-$ and $H^{13}CO_3^-$. In Exp. 3, specimens of *O. universa* were maintained in ambient seawater and fed small pieces of $^{15}N$- and $^{13}C$-enriched *A. salina*, and then followed for eight additional hours (figure 1).

Experiments 1 and 2 were initiated at 13.00 local time (corresponding to the onset of the period of maximum symbiont photosynthetic rate [2]), which we refer to as $t = 0$ h. Twenty-four specimens were transferred into 22 ml borosilicate glass scintillation vials (one specimen per vial) which were filled with 0.45 µm filtered seawater spiked by the addition of 2 mM $NaH^{13}CO_3$ (99% $^{13}C$; Cambridge Isotopes Laboratory Inc.) and either 10 µM $^{15}NH_4Cl$ (99% $^{15}N$; Cambridge Isotopes Laboratory Inc.) for Exp. 1 (12 specimens) or 10 µM $K^{15}NO_3$ (99% $^{15}N$; Cambridge Isotopes Laboratory Inc.) for Exp. 2 (12 specimens). The pH of the seawater after the addition of the isotope salts was 8.2. Addition of the spike resulted in a final dissolved inorganic carbon (DIC) concentration of approximately 4 mM and a DIC $^{13}C/^{12}C$ ratio of approximately 0.45. San Pedro basin surface water concentrations of $NO_3$ are assumed to be zero, and around 0.2 µM for $NH_4$ [17].

For Exp. 3, three specimens were transferred into 22 ml scintillation vials (one specimen per vial) which was filled with 0.45 µm filtered seawater, and fed a small piece of $^{15}N$- and $^{13}C$-enriched *A. salina* that had been thawed from the freezer at 11.00 local time ($t' = 0$ h for this experiment). The *Artemia* pieces were gently pipetted onto the *O. universa* spines at which time the foraminifera transported the food particles to the surface of the shell where it

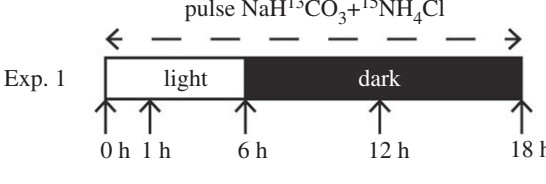

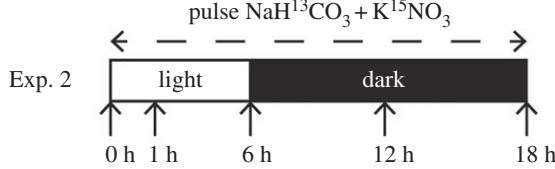

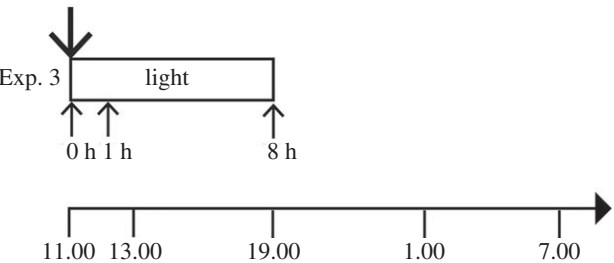

**Figure 1.** *Orbulina universa* incubation experiment timelines. Three specimens were collected and fixed for TEM/NanoSIMS analyses at each time point (lower arrows). Details in the text.

ingested the food via the production of food vacuoles (see https://www.youtube.com/watch?v=BYQNt52tiVU). The three specimens used in this experiment trapped the food particles in their spines within a few minutes.

Following specimen transfer into labelled solutions or following labelled *Artemia* feeding, the vials were capped and immersed in water baths at 22°C under artificial light (Sylvania F24T12 'Cool White' fluorescent lights). Irradiance levels exceeded 350 µmol photons $m^{-2}$ $s^{-1}$, which is the $P_{max}$ light saturation threshold for symbiont photosynthesis in this species [2,18]. Specimens were maintained on a 12/12 h light/dark cycle, similar to natural conditions, with lights off at 19.00 local time (dark period), and lights back on at 07.00 the next day. At pre-defined sampling times ($t$ = 1 h, 6 h, 12 h, or 18 h for experiments 1 and 2, and $t$ = 8 h for Exp. 3 (figure 1)), three specimens were moved into vials for TEM fixation. Three additional individuals were maintained in unspiked filtered seawater and fixed before the experiments to serve as unlabelled control specimens.

### (d) Chemical fixation and TEM observation

At each sampling time point, selected *O. universa* specimens were individually transferred into 0.5 ml micro-centrifuge tubes and processed exactly as described by LeKieffre *et al.* [15] (see electronic supplementary material, text S2 for details).

### (e) NanoSIMS analysis

NanoSIMS settings and analysis procedures follow that described by LeKieffre *et al.* [15,19] (see electronic supplementary material, text S2 for details), except that both $^{13/12}$C and $^{15/14}$N ratios were measured. Quantified $^{13}C/^{12}C$ and $^{15}N/^{14}N$ ratios were obtained as follows:

$$\delta^{13}C\ (‰) = \left( \left( \frac{^{13/12}C_{meas}}{^{13/12}C_{control}} \right) - 1 \right) \times 10^3$$

and

$$\delta^{15}N(‰) = \left( \left( \frac{^{15/14}N_{meas}}{^{15/14}N_{control}} \right) - 1 \right) \times 10^3.$$

NanoSIMS image processing was carried out as described in LeKieffre *et al.* [15,19] using the software Look@NanoSIMS [20] (see electronic supplementary material, text S2 for details). Isotopic enrichments were calculated for the foraminifer cytoplasm, dinoflagellate, and cellular compartments. For each condition, time point, and type of cellular compartments, the average $^{13}$C- and $^{15}$N-enrichments were calculated based on three replicates (unless otherwise specified).

### (f) Statistics

Statistical analysis was performed as described in [15] (see electronic supplementary material, text S3 for details). Briefly, statistical analysis was carried out on the total set of ROIs (regions of interest) for each time point using a linear mixed-effect model taking into account pseudo-replication effects, followed by a Tukey multiple comparison test. The results of the linear mixed-effect model are listed in the electronic supplementary material, text S4. Comparisons of $\delta^{13}$C of the different cell compartments between Exp. 1 and Exp. 2 were performed with $t$-tests for each cell compartment and each time point. The results of the $t$-tests are listed in the electronic supplementary material, table S2. All statistical analysis were performed with the RStudio software [21] with the significance level set to $\alpha$ = 0.05 (i.e. $p$-value < 0.05).

## 3. Results

### (a) Ultrastructure observations

All *O. universa* observed in this study exhibit the same ultrastructural features described previously [15,22–24]. Specimens have symbiotic dinoflagellates containing a nucleus with condensed chromosomes, two peripheral chloroplasts with stalked pyrenoids surrounded by starch, and accumulated starch grains within the cell cytosol for symbiotic dinoflagellates sampled at the end of the light phase ($t$ = 6 h), whereas vacuolarized dinoflagellates in the foraminifer endoplasm at the end of the dark phase ($t$ = 18 h) were nearly starch-free (figure 2 and electronic supplementary material, figures S1 and S2).

### (b) Inorganic $^{13}$C-assimilation

For both Exp. 1 and Exp. 2, the NanoSIMS images show clear $^{13}$C-accumulation in the symbiotic dinoflagellates and foraminiferal cytoplasm (figure 3 and electronic supplementary material, figure S2). No qualitative or statistically quantitative differences were seen in the $^{13}$C-assimilation between the Exp. 1 and Exp. 2, except at the first time point, $t$ = 1 h. At this time point, elevated $^{13}$C-enrichment was found in the lipid droplets from individuals incubated in Exp. 1 and in the dinoflagellates from individuals incubated in Exp. 2 (figure 3). Apart from these observations, the $^{13}$C-enrichments in the two experiments were found in the same organelles and at similar levels (figure 3 and electronic supplementary material, table S2).

Accumulation of $^{13}$C-labelled C is first seen in the dinoflagellate starch surrounding the chloroplast pyrenoids (electronic supplementary material, figure S2) after 1 h of incubation. After 6 h of incubation (at the end of the light phase), numerous $^{13}$C-labelled starch grains can be seen in the dinoflagellate symbiont cells, with $\delta^{13}$C values of approximately 10 000–40 000‰. At this time, $^{13}$C-enrichment is also observed in organelles throughout the dinoflagellate cell. The $^{13}$C-enrichment levels remain stable during the first 6 h of the dark phase (between $t$ = 6 and $t$ = 12 h; figure 3). By the end of the dark phase at $t$ = 18 h, the $\delta^{13}$C content of the dinoflagellates decreases to approximately 3000‰ due to the near complete

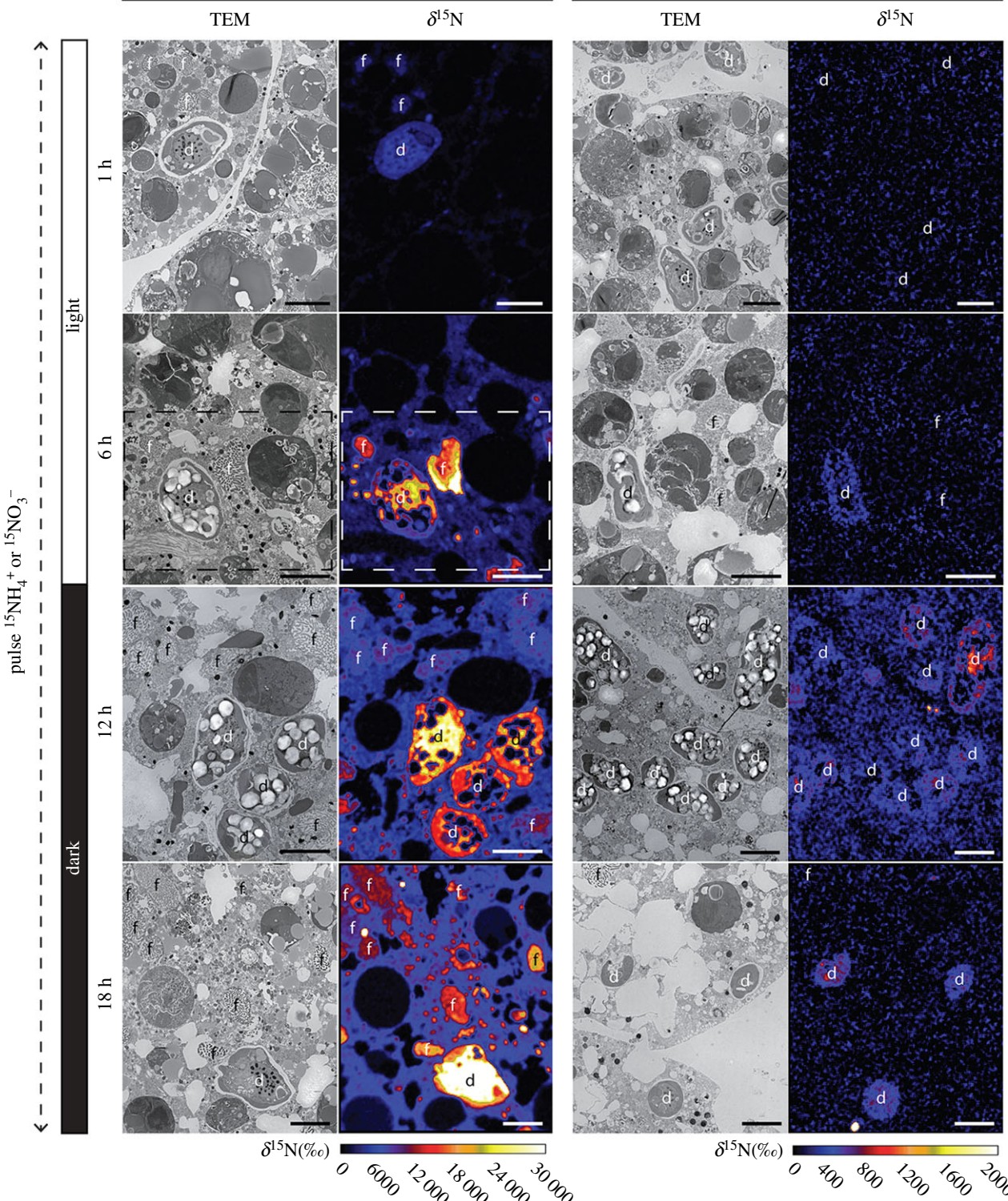

**Figure 2.** Time-evolution of $^{15}N$ incorporation into symbiotic dinoflagellates and *Orbulina universa* organelles and cytoplasm during the light phase ($t = 1$ and 6 h), and dark phase ($t = 12$ and 18 h). TEM micrographs of *O. universa* endoplasm with symbiotic dinoflagellates are adjacent to the corresponding NanoSIMS images that show the $^{15}N/^{14}N$ distribution across the TEM image (expressed as $\delta^{15}N$ in ‰). Left 2 columns display incubations with $^{15}NH_4^+$ from Exp. 1. Right 2 columns show incubations with $^{15}NO_3^-$ from Exp. 2. The boxed region in Exp. 1 $^{15}NH_4$, $t = 6$ h images are enlarged in electronic supplementary material, figure S4. d: dinoflagellate, f: fibrillar body. Note the NanoSIMS scales for the $^{15}NH_4^+$ and $^{15}NO_3^-$ incubations differ by a factor of 15. Scale bars: 5 µm. (Online version in colour.)

utilization of stored starch and translocation of carbon to the foraminifera host cytoplasm.

In the foraminiferal cell, a weak $^{13}C$-enrichment can be seen in the foraminiferal lipid droplets starting at $t = 1$ h (figure 3 and electronic supplementary material, figure S2). After 6 h in light, the foraminiferal endoplasm exhibits $^{13}C$-labelled hot-spots, mostly in lipid droplets, electron-opaque bodies, and the

fibrillar bodies. $\delta^{13}C$ values approach 1300‰, 800‰, and 400‰, respectively. NanoSIMS images taken on specimens from the dark phase at $t = 12$ h reveal a greater proportion of $^{13}C$-enriched lipid droplets than during the light phase of the experiment ($t = 1$–6 h). Here, lipid droplets reach values between 1500 and 2500‰. It is noteworthy that dinoflagellate lipids and the lipid droplets observed along their periphery or

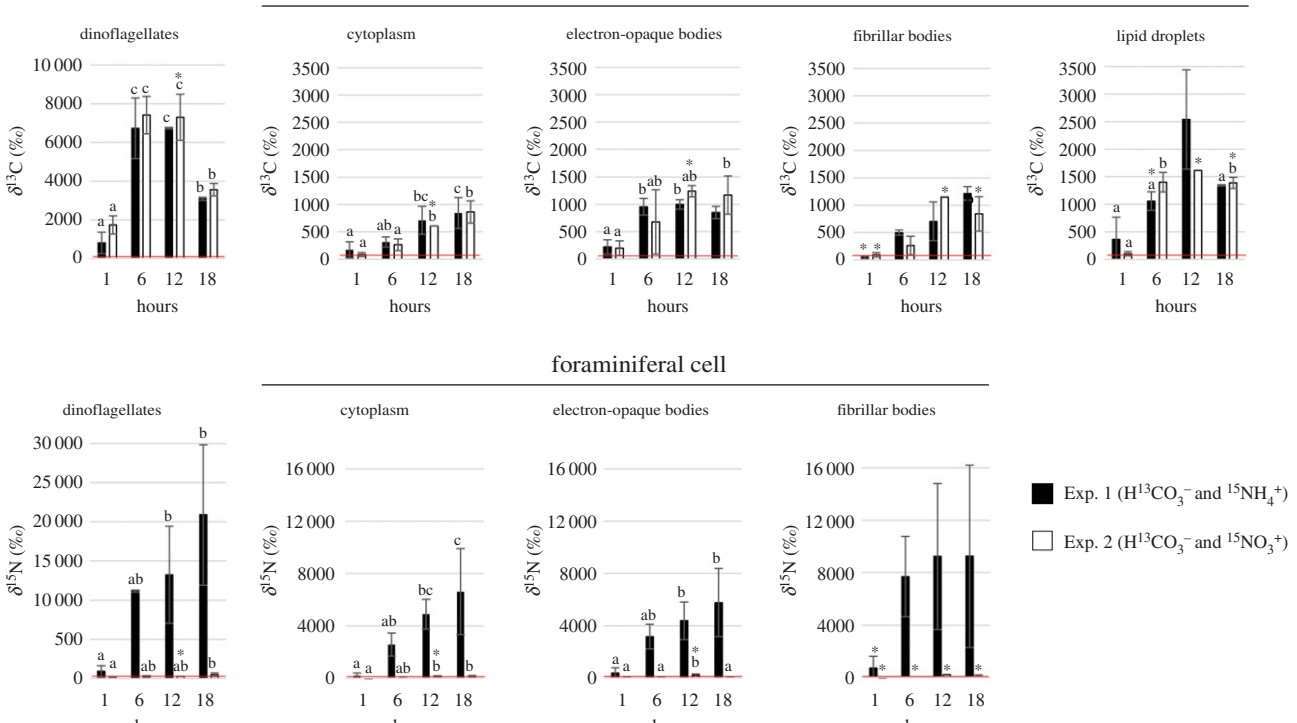

**Figure 3.** Average $\delta^{13}C$ and $\delta^{15}N$ values in different cell compartments of *Orbulina universa* incubated with $H^{13}CO_3^-$ and $^{15}NH_4^+$ (Exp. 1, black bars) or with $H^{13}CO_3^-$ and $^{15}NO_3^-$ (Exp. 2, white bars). Measurements were made for symbiotic dinoflagellates, foraminiferal cytoplasm, fibrillar bodies, electron-opaque bodies, and lipid droplets. Error bars represent one standard deviation ($n = 3$, except values marked with an asterisk: $n = 2$). Baseline $\delta^{13}C$ and $\delta^{15}N$ values for the NanoSIMS control specimens that were grown in unlabelled seawater are $\delta^{13}C = 0 \pm 50‰$ and $\delta^{15}N = 0 \pm 51‰$, ($n = 2; \pm 3\sigma$).

in the region between the dinoflagellate and the symbiosome displayed similar $^{13}C$-enrichment values (electronic supplementary material, figure S3). The electron-opaque body (small electron-dense circular vesicle of approx. 500 nm diameter) $\delta^{13}C$ remain stable from $t = 6$ to 18 h (through the dark phase). By contrast, the $^{13}C$-enrichment of the foraminiferal cytoplasm increases significantly between $t = 6$ and 18 h, reaching approximately 850‰. Protein-rich fibrillar bodies [25,26] are found throughout the cytoplasm (electronic supplementary material, figure S2), and a $^{13}C$-enrichment in some, but not all of them could also be detected after $t = 6$ h with values ranging between 300 and 500‰ (figure 3). In fibrillar body structures displaying $^{13}C$ uptake, the $^{13}C$-enrichment appears to increase during the first period of the dark phase ($t = 6$–12 h), and then remains stable between $t = 12$ and 18 h with values ranging from 700 to 1200‰.

## (c) Inorganic $^{15}N$-assimilation

NanoSIMS data reveal a significantly greater $^{15}N$-assimilation rate when *O. universa* and its symbiotic dinoflagellates are incubated with $^{15}NH_4^+$ (Exp. 1) than when they are incubated with $^{15}NO_3^-$ (Exp. 2). Regions of $^{15}N$-enrichment are detected in the dinoflagellates after only 1 h exposure to $^{15}NH_4^+$, whereas $^{15}N$-enrichment is not observed in the dinoflagellates until after 6 h of exposure to $^{15}NO_3^-$ in the same concentration (figure 2). In both experiments, the $^{15}N$-enrichment in the dinoflagellates progressively increases during light and dark phases, but the enrichment level for $^{15}NH_4^+$ in Exp. 1 is 40 times greater than seen for $^{15}NO_3^-$ in Exp. 2: approximately 21 000‰ versus approximately 500‰ at the end of the dark phase, respectively (figures 2 and 3 at $t = 18$ h). All the cellular compartments of the dinoflagellates are $^{15}N$-enriched except the starch grains and lipid droplets which lack nitrogen (figure 2,

details in electronic supplementary material, figure S4). Similar to the enrichment observed in the symbionts, the $^{15}N$-enrichment in the foraminiferal cytoplasm and electron-opaque bodies increases steadily throughout the Exp. 1 $^{15}NH_4^+$ incubation period attaining values of approximately 6500‰ and approximately 6000‰, respectively, by $t = 18$ h (figures 2 and 3). Most of the fibrillar bodies observed in the Exp. 1 images were enriched in $^{15}N$, with labelled values exceeding the $^{15}N$-enrichment of the surrounding endoplasm (figures 2 and 3). $^{15}N$-enrichment values of fibrillar bodies increased from 700 to 9500‰ between the first observation of labelled organelles at $t = 1$ h and $t = 18$ h (figure 3). At higher magnification, highly enriched $^{15}N$-labelled vesicles (approx. 20 000‰) were observed in the foraminiferal endoplasm and along the periphery of the dinoflagellates (figure 2, details in electronic supplementary material, figure S4). In Exp. 2 ($^{15}NO_3^-$ incubation), only the dinoflagellates and the foraminiferal fibrillar bodies showed $^{15}N$-enrichment with values reaching a maximum of approximately 700‰ and approximately 750‰, respectively (figures 2 and 3). At the end of the incubation after 18 h, the $^{15}N$ values of the foraminiferal cytoplasm in Exp. 2 (ca 130‰) were only slightly higher than the level recorded in control specimens ($0 \pm 51‰$, $3\sigma$). However, in the three *O. universa* specimens incubated for 18 h in Exp. 2, we observe small vesicles (0.5–1 µm diameter) that were noticeably enriched in $^{15}N$ with $\delta^{15}N$ values of approximately 30 000‰ (electronic supplementary material, figure S5). These vesicles, potentially prokaryotes at different states of degradation, were always enclosed in what appeared to be degradation vacuoles.

## (d) Feeding: organic $^{13}C$- and $^{15}N$-assimilation

TEM observations reveal that the cytoplasm of the *O. universa* specimens fixed 8 h after being fed with $^{13}C$- and $^{15}N$-enriched

*Artemia* fragments (Exp. 3) is filled with numerous digestive vacuoles (figure 4) that are characterized by an ovoid to circular shape, and a diameter ranging from 1 to 5 μm. These digestive vacuoles display both $^{13}$C- and $^{15}$N-labelling (figure 4), confirming that their content is $^{13}$C- and $^{15}$N-enriched fragments of *Artemia*; the only source of $^{13}$C and $^{15}$N in this experiment. The enrichment levels were different between the two individuals analysed: with average $\delta^{13}$C values of approximately 200‰ and approximately 1300‰, and average $\delta^{15}$N values of approximately 2000‰ and approximately 45 000‰, respectively (table 1). The dinoflagellates present in *O. universa* cytoplasm also exhibited significant $^{15}$N-enrichment, with values ranging from 300 to more than 6000‰ (table 1 and figure 4). However, their $^{13}$C-enrichments were similar to the unlabelled control specimens.

## 4. Discussion

### (a) Carbon assimilation by the symbionts and translocation to the host cell

The dynamics of photosynthetic C-fixation and starch accumulation by symbiotic dinoflagellates and subsequent transfer of organic C to the *O. universa* cell were described in a previous NanoSIMS pulse-chase experiment [15]. These authors reported that within 45 min, $^{13}$C-labelled starch accumulated around the chloroplast pyrenoids and by the end of the 6-h pulse phase, numerous $^{13}$C-labelled starch granules could be observed within the dinoflagellate cytosol. During the dark phase, $^{13}$C-labelled starch was converted to lipid in the symbiont cell and then transferred to the foraminifera cytoplasm, where it was assimilated and distributed throughout the cytoplasm. The present work supports these previous observations (figure 3 and electronic supplementary material, figure S2) and the pathway for carbon translocation from symbionts to planktonic foraminifera, which has been estimated to contribute about 40% of the total C budget of the host-symbiont complex [1]. In other symbiotic associations involving dinoflagellates, carbon translocation to the host occurs through the movement of compounds such as glycerol, glucose, lipids, organic acids, or amino acids [27–30]. However, the study of the holobiont transcriptome in symbiotic radiolarians, another marine planktonic protozoan, revealed that most of the microalgal genes involved in glucose, fatty acids, or glycerol production were not upregulated, suggesting that amino acids might be the major form of C translocation from symbionts to the host [31]. In the present study, small lipid droplets are seen in the foraminiferal cytoplasm in direct contact with the symbiosome, in between the symbiosome and the dinoflagellate membrane, and within the dinoflagellate cytosol during the dark phase (electronic supplementary material, figure S3). These observations further support our hypothesis that lipids are an important vehicle for C transfer across the dinoflagellate and symbiosome membrane in planktonic foraminifera, but does not exclude the possibility that a concomitant transfer of C also occurs in soluble form, e.g. as amino acids.

The slightly higher $^{13}$C-enrichment observed after 1 h in the dinoflagellates incubated in Exp. 2 compared to those from Exp. 1 could result from different metabolic pathways for metabolite release. Dinoflagellates incubated in Exp. 1 had access to excess ammonium, while those incubated in Exp. 2 only had access to nitrate, which they did not assimilate (see further in the Discussion). When deprived of a usable nitrate

source, dinoflagellates switch their carbon metabolism to more lipid production rather than N-containing compounds (e.g. amino acids and therefore proteins, nucleic acids) [32]. A similar pathway in the *O. universa* symbiotic dinoflagellates could explain the higher $^{13}$C-enrichment observed in symbionts from Exp. 2 (accumulation of the $^{13}$C in their starch granules and/or lipid droplets) while in Exp. 1, more soluble N-compounds (e.g. amino acids) could have been produced and lost during sample preparation.

Note that the $^{13}$C-enrichment differences observed at $t = 1$ h for the lipid droplets between the Exp. 1 and the Exp. 2 specimens could be due to high inter-individual variability, as one of the three specimens incubated in Exp. 1 exhibited much higher $^{13}$C-values than the two other specimens. Therefore, the analysis of a larger number of specimens is recommended for future experiments.

### (b) Inorganic nitrogen assimilation and exchanges between the symbionts and the host cell

One of the robust observations from this study is that the symbiotic dinoflagellates display a clear preference for ammonium over nitrate assimilation. Ammonium-$^{15}$N accumulation was up to 40 times higher than nitrate accumulation after 18 h of incubation (figures 2 and 3). The $^{15}$N-enrichment in all the foraminiferal compartments investigated (cytoplasm, electron-opaque, and fibrillar bodies) was much higher in Exp. 1 ($^{15}$NH$_4^+$) compared to Exp. 2 ($^{15}$NO$_3^-$). In Exp. 1, ammonium assimilation into the dinoflagellate nucleus and chloroplasts as well as the protein-rich foraminiferal fibrillar bodies (figure 2) was visible already after 1 h of incubation. By $t = 6$ h, the $^{15}$N-tracer was distributed throughout the foraminiferal cytoplasm, but was more concentrated in the symbiont and foraminifera organelles associated with protein and nucleotide synthesis, in addition to small electron-opaque bodies (figures 2 and 3). In all the cell compartments investigated including the dinoflagellates, the $^{15}$N concentration increased throughout the duration of the experiment (light and dark phases). It is noteworthy that, contrary to inorganic carbon where only the photosynthetic dinoflagellates are capable of C-assimilation [15], ammonium could be assimilated directly by the foraminiferal cell. Ammonium assimilation via a cytoplasmic pathway was suggested for benthic kleptoplast (i.e. stolen chloroplasts) species from aphotic habitats, based on the observation of ammonium assimilation in the dark while the kleptoplasts were photosynthetically inactive [9]. Thus, the increase of $^{15}$N-enrichment in the foraminiferal compartments observed over the 18 h incubation likely results from a combination of a transfer of $^{15}$N-compounds from the dinoflagellates to the foraminiferal host cell and assimilation of $^{15}$N-ammonium by one or more foraminiferal cytoplasmic pathways.

Generally, photosynthetic organisms such as algae display a tight coupling of inorganic C and N uptake, with photosynthesis providing the carbon skeleton necessary for N assimilation [33]. In our experiment, $\delta^{15}$N increased in the dinoflagellate cytoplasm during both the light and dark phases, the foraminifera symbionts are capable of assimilating ammonium N throughout the entire diurnal cycle examined. Dark assimilation of ammonium has been reported previously in different dinoflagellate species [32], and dinoflagellate symbionts in corals assimilate nitrate in the dark even after corals were acclimated to darkness for 24 h [34]. These observations demonstrate the ability of symbiotic dinoflagellates to

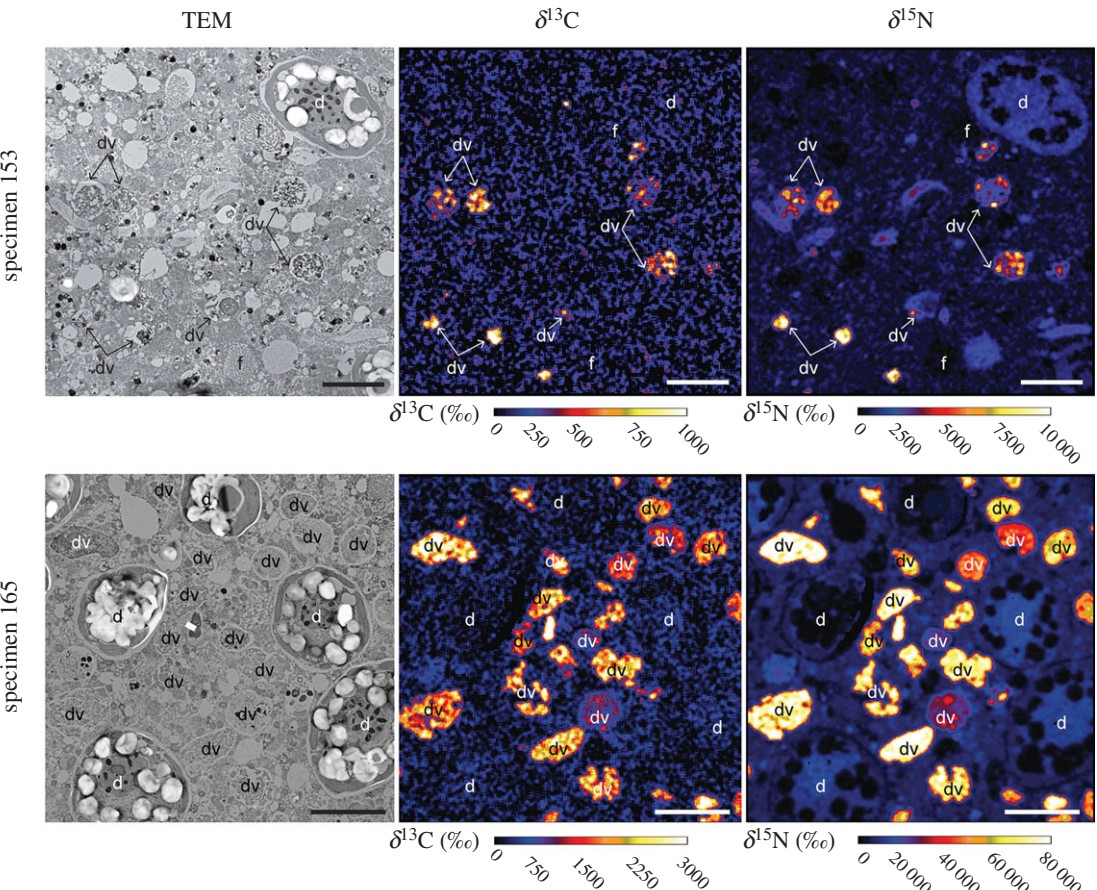

**Figure 4.** TEM and NanoSIMS images of cytoplasm from two *Orbulina universa* specimens at $t = 8$ h in feeding Exp. 3. The bright $^{13}$C- and $^{15}$N-enriched regions are digestive vacuoles containing tissue from $^{13}$C-$^{15}$N-labelled *Artemia salina* food that was fed to the foraminifers earlier (expressed as $\delta^{13}$C and $\delta^{15}$N in ‰). The NanoSIMS images show that digested $^{15}$N is distributed throughout the cytoplasm endoplasm by $t = 8$ h. d: dinoflagellate, dv: digestive vacuole, f: fibrillar body. Scale bars: 2 µm. (Online version in colour.)

**Table 1.** Feeding experiment (Exp. 3): $\delta^{13}$C and $\delta^{15}$N values in the digestive vacuoles observed in the cytoplasm of the two *O. universa* specimens incubated 8 h after feeding with a particle of $^{13}$C- and $^{15}$N-labelled *Artemia nauplii*.

| | digestive vacuoles | | dinoflagellates | | |
|---|---|---|---|---|---|
| | $\delta^{13}$C (‰) | $\delta^{15}$N (‰) | $\delta^{13}$C (‰) | $\delta^{15}$N (‰) | food particles on foraminiferal spines after 3 h |
| Specimen 153 | 211 ± 116 | 1922 ± 1079 | 24 ± 20 | 389 ± 291 | no |
| Specimen 165 | 1287 ± 383 | 45468 ± 15295 | 132 ± 53 | 6069 ± 1970 | yes |

assimilate N during the night in the absence of carbon skeletons supplied by photosynthetic processes. Although corals store nitrogen in highly N-concentrated uric acid crystals within the dinoflagellate cells, [34], we have not observed such crystal structures within the dinoflagellate cytoplasm in *O. universa* (figure 3 and electronic supplementary material, figure S4). Nevertheless, we note that dinoflagellates are known to be able to store large amounts of N within their cells, potentially in the form of free amino acids [32], which in our case would have been lost during sample preparation.

## (c) Nitrate assimilation inhibited by ammonium in symbiotic microalgae

*Orbulina universa* has a wide habitat range, ranging from the tropics through the oligotrophic gyres to the subtropical convergences and equatorial and coastal upwelling zones. This species complex has the broadest temperature tolerance among extant planktonic foraminifera species with a range

from 9 to 30°C [35,36]. In oligotrophic regions, ammonium is generally scarce but is thought to serve as the main source of N for phytoplankton because of its high turnover rate [37–39].

Ammonium assimilation results in a better energy yield than nitrate assimilation [40] and thus nitrate uptake is inhibited with increasing ammonium concentration in photosynthetic organisms [41–43], including dinoflagellates [44]. Higher assimilation of ammonium over nitrate has been observed in other photosynthetic organisms such as cyanobacteria (*Prochlorococcus* and *Synechococcus*) [45]. The gene expression involved in N assimilation depends on the primary form of N that is available to the photosynthetic organisms [46–48]. In *Thalassicolla nucleata*, a planktonic radiolarian found in association with the photosynthetic symbiotic dinoflagellate *Brandtodinium* sp., holobiont transcriptome analysis revealed a dramatic downregulation of the genes involved in nitrate reduction while the genes involved in ammonium assimilation were all upregulated [31].

The $NO_3$ half-inhibition concentration of $NH_4$ was estimated for two dinoflagellate species at around 1.3 µM [49],

so we assume that the natural $NH_4$ concentration measured in the surface waters of San Pedro basin (0.2 µM) is not sufficient to inhibit $NO_3$ assimilation by the symbiotic dinoflagellates. Rather, we suggest that in the *O. universa* symbiotic system, the host foraminifer provides sufficient recycled ammonium to its dinoflagellates such that similar gene expression regulation likely occurs which would result in the inhibition of the nitrate assimilation pathway. Indeed, in our experiments, all the foraminifera were fed the day before the experiment, and the feeding experiment (Exp. 3) showed that prey were still being digested 6 h after the feeding event. Thus, it is likely that recycled ammonium was still available to the symbionts and host in the foraminifera incubated in Experiment 2 ($^{15}NO_3^-$). We also cannot exclude the possibility that the foraminifera might be able to excrete ammonium from N-stored pools. Nevertheless, the ability of the *O. universa* dinoflagellates to efficiently use ammonium N whenever available is a powerful evolutionary adaptation for this protist association to tightly recycle bioavailable nitrogen in oligotrophic environments, regardless of its source from food or dissolved inorganic N.

### (d) Nitrate assimilation via feeding on prokaryotes?

Despite the fact that *O. universa* was incubated in filtered (0.45 µm) ambient seawater, we could not avoid incidental bacterial contamination or prevent prokaryotes from being introduced into the culture vessel, if they were not already present on the foraminiferal spines and rhizopodia. Indeed, in Exp. 2 ($^{15}NO_3^-$), the highly $^{15}N$-enriched vesicles we observe in the *O. universa* cytoplasm (electronic supplementary material, figure S5) have a size range between 0.5 and 1 µm and could therefore be prokaryotes. As these vesicles were in various states of degradation and always enclosed within degradation vacuoles, it is unlikely that they are endosymbionts. These prokaryote-like vesicles could have assimilated $^{15}NO_3^-$ from the seawater while residing outside the foraminiferal cytoplasm in the spines or rhizopodia. In this case, they could have been ingested by the foraminifer after the experiment began. The digestion of these $^{15}N$-enriched prokaryote-like vesicles could result in the production of recycled $^{15}N$-compounds that, in turn, were either directly transferred to the dinoflagellate, or excreted in their microenvironment and reacquired by the symbionts. This process does not necessary involve a remineralization step into ammonium as symbiotic dinoflagellates are also able to assimilate organic nitrogen (dissolved amino acids, urea) [30,34,50]. Thus, we suggest that the weak $^{15}N$-nitrate assimilation observed in the foraminiferal cell and its dinoflagellates could also be the result of digestion of these $^{15}N$-labelled vesicles (potential prokaryotes) rather than exclusively through direct nitrate uptake by the symbiotic dinoflagellates.

### (e) Nitrogen assimilation derived from prey capture by the foraminiferal host

The two foraminifera fed with $^{13}C$- and $^{15}N$-labelled *Artemia* (Exp. 3) contain cytoplasm filled with digestive vacuoles (figure 4). The lower $^{13}C$- and $^{15}N$ enrichment values observed for specimen 153 could be due to a lower enrichment of the $^{13}C$- and $^{15}N$-labelled *Artemia* particle it was fed. Indeed, $^{13}C$- and $^{15}N$-enrichments were measured on a lyophilized aliquot, but did not take into account enrichment differences between individuals or tissue components [16]. Regardless, in both

specimens the symbiotic dinoflagellates were found labelled with $^{15}N$ after 8 h. Since all the $^{13}C$- and $^{15}N$-labelling in this experiment must come from prey digestion, the observation of $^{15}N$-enriched dinoflagellates in the foraminiferal cytoplasm attests to the transfer of $^{15}N$-compounds between the digestive vacuoles and the symbiotic dinoflagellates. This transfer could happen via a direct intracellular transfer of N-compounds from the foraminifer to the dinoflagellate, or more likely, via symbiont assimilation of recycled ammonium excreted into the extracellular microenvironment of the foraminifera spines since the symbionts were primarily on the spines and not in vacuoles within the foraminifera cytoplasm during this experiment. Additional experiments are required to more completely document the diurnal dynamics of foraminiferal food-derived N assimilation, ammonium remineralization, and subsequent excretion/uptake by the symbiotic dinoflagellates. However, we show here that symbiotic dinoflagellates do use recycled N derived from food digestion by the host. These observations confirm the model proposed by Uhle *et al.* [5] that most of the nitrogen used by the *O. universa* symbionts comes from recycled ammonium excreted by the foraminifera, rather than direct nitrate assimilation by the dinoflagellates. This mechanism is likely used by other species of dinoflagellate bearing planktonic foraminifera and could be common among other symbiotic systems as it was also observed in other symbioses (e.g. cnidarians, sponges; e.g. [51–53]). In coral symbiosis, studies estimated that symbiotic dinoflagellates could meet up to 90% of their N requirements through N recycling [52,53].

### (f) Fibrillar bodies: protein storage organelles for biomineralization

The results from Exp. 1 (figures 2 and 3) show that the foraminifera organelles most actively synthesizing proteins during the day are the fibrillar bodies that make up the fibrillar system [54]. These enigmatic organelles are only found in planktonic foraminifera and were first described by Rhumbler [55] who called them 'Gallertstränge' (meaning gelatinous strands). Later, light [25] and TEM [26,56] examination of planktonic foraminifera ultrastructure reported that the fibrillar bodies were composed of strands of proteins that were located primarily in the final and penultimate chambers. Hansen [26] first speculated that the fibrillar system could be linked to foraminifer buoyancy although no evidence was presented to support this hypothesis. Subsequent TEM examination of *Orbulina universa* ultrastructure during chamber formation [57] documented the presence of fibrillar material in the extracellular environment adjacent to a newly forming chamber, suggesting the fibrillar material is linked to biomineralization during initial chamber formation rather than serving a buoyancy function.

Active uptake of $^{15}NH_4$ by the fibrillar bodies in Exp. 1 is intriguing in that the rapid synthesis and accumulation of protein during the day must serve a fundamental and time-sensitive role in foraminifera physiology for individuals to expend so much energy in protein synthesis. Electronic supplementary material, figure S1 also shows that the fibrillar bodies are incorporating $^{13}C$ on a timescale similar to that of $^{15}N$. Chamber formation in planktonic foraminifera is an episodic event that involves the rapid transport and organization of a polysaccharide-protein-rich organic matrix outside the foraminifera endoplasm [58]. The entire process from initial organic matrix extrusion through initial chamber wall biomineralization occurs in 1–2 h in *O. universa* [57]. Hence, the

biomineralizing proteins and associated polysaccharides needed for chamber formation must be synthesized and stored within the foraminifera cell prior to a biomineralization event in order for chamber formation to occur so rapidly. We suggest that the fibrillar system is the synthesis and storage organelle for the organic matrix proteins/ polysaccharides prior to chamber biomineralization.

## 5. Conclusion

Our data show that ammonium is the major source of N used for protein and nuclear synthesis by the symbiotic dinoflagellates in the planktonic foraminifera *Orbulina universa*. The observation of $^{15}$N-labelled dinoflagellate cells after feeding the foraminifera with $^{15}$N-*Artemia* shows that symbiotic dinoflagellates use recycled ammonium (or other N-compounds). As *O. universa* typically inhabits oligotrophic waters where ammonium is scarce, recycled ammonium provided by the host is likely a key source of nitrogen for protein and nucleotide synthesis in the symbiotic dinoflagellates. These findings are of major interest for the interpretation of foraminifera-bound $^{15}$N/$^{14}$N ratios from fossil shells (FB-$\delta^{15}$N), as recycled ammonium does not necessarily have the same isotopic fractionation that other nitrogen sources exhibit such as nitrate. Considering an isotopic balance within the symbiont-host system, complete uptake of recycled ammonium requires the recycled ammonium to have the same $^{15}$N/$^{14}$N ratios as the external N source, e.g. the prey, yet any ammonium excretion due to incomplete uptake will cause an isotopic difference between recycled ammonium and the organic matter produced. Our evidence thus raises an interesting mechanism for interpreting the observed species FB-$\delta^{15}$N difference in fossil records. Specifically, it is possible that the non-symbiotic foraminifera, or foraminifera species bearing non-dinoflagellate symbionts, may exhibit more elevated $\delta^{15}$N in their biomass and shell-bound organic N, because of the lack of efficient N recycling processes observed between dinoflagellate symbionts and hosts.

Data accessibility. The datasets generated during and/or analysed during the current study are available in the Zenodo repository (doi:10.5281/zenodo.3834220).

Competing interests. We declare we have no competing interests.

Funding. This work was supported by the Swiss National Science Foundation (grant no. 200021_149333) and the US National Science Foundation (grant no. OCE-1261516 to J.S.F. and A.D.R.) and grant no. OCE-0550703 (H.J.S.).

Acknowledgement. We gratefully acknowledge the staff of the University of Southern California, Wrigley Marine Science Center for field and laboratory assistance. We thank Team Catalina (O. Branson, T. Bergamaschi, E. Bonnin, E. Chu, C. Livsey, C. Pope, E. Schoenig, K. Davis, and J. Snyder) for their skilled participation in the laboratory and offshore foraminifera collection. The electron microscopy platform at the University of Lausanne (Switzerland) is thanked for advice and for access to the equipment. Dr Stéphane Escrig is thanked for expertise help with NanoSIMS imaging.

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
