## [Reviewer comments · Proceedings of the Royal Society B: Biological Sciences]

Review History

RSPB-2020-0620.R0 (Original submission)

Review form: Reviewer 1

Recommendation

Accept as is

Scientific importance: Is the manuscript an original and important contribution to its field?
Excellent

General interest: Is the paper of sufficient general interest?
Good

Quality of the paper: Is the overall quality of the paper suitable?
Acceptable

Is the length of the paper justified?
Yes

Should the paper be seen by a specialist statistical reviewer?
No

Do you have any concerns about statistical analyses in this paper? If so, please specify them explicitly in your report.

Yes

It is a condition of publication that authors make their supporting data, code and materials available - either as supplementary material or hosted in an external repository. Please rate, if applicable, the supporting data on the following criteria.

Is it accessible?

Yes

Is it clear?

Yes

Is it adequate?

N/A

Do you have any ethical concerns with this paper?

Yes

Comments to the Author

The target is clear, experiment was well designed, and the results are also clear and informative.

Review form: Reviewer 2

Recommendation

Accept with minor revision (please list in comments)

Scientific importance: Is the manuscript an original and important contribution to its field?

Excellent

General interest: Is the paper of sufficient general interest?

Excellent

Quality of the paper: Is the overall quality of the paper suitable?

Excellent

Is the length of the paper justified?

Yes

Should the paper be seen by a specialist statistical reviewer?

Yes

Do you have any concerns about statistical analyses in this paper? If so, please specify them explicitly in your report.

No

It is a condition of publication that authors make their supporting data, code and materials available - either as supplementary material or hosted in an external repository. Please rate, if applicable, the supporting data on the following criteria.

Is it accessible?

Yes

Is it clear?

Yes

Is it adequate?

Yes

Do you have any ethical concerns with this paper?

No

Comments to the Author

I only have a couple of minor editorial suggestions:

- 1) In the introduction, I would clarify (or rather, elaborate in a bit more details) the role of foraminifera as a component of primary production: while they do rely on heterotrophy for nitrogen, they are part of PP due to carbon fixation by symbionts
- 2) be consistent in presenting number formats (e.g. lines 261 and 270)
- 3) change to "highly ^{15}N -enriched..." (line 388)
- 4) I would like to see a sentence or two in the conclusions addressing the difference in $\delta^{15}\text{N}$ between dinoflagellate-bearing foraminifera and those with pelagophyte symbionts, which was brought up in the intro (Lines 67-69). In light of your findings, can you speculate what could drive the difference?
- 5) It is interesting that metabolic products from symbionts have higher isotopic values than the diet (lines 55-56). I think it might warrant a sentence or two of explanation.

Review form: Reviewer 3

Recommendation

Accept with minor revision (please list in comments)

Scientific importance: Is the manuscript an original and important contribution to its field?

Excellent

General interest: Is the paper of sufficient general interest?

Good

Quality of the paper: Is the overall quality of the paper suitable?

Excellent

Is the length of the paper justified?

Yes

Should the paper be seen by a specialist statistical reviewer?

No

Do you have any concerns about statistical analyses in this paper? If so, please specify them explicitly in your report.

No

It is a condition of publication that authors make their supporting data, code and materials available - either as supplementary material or hosted in an external repository. Please rate, if applicable, the supporting data on the following criteria.

Is it accessible?

Yes

Is it clear?

Yes

Is it adequate?

Yes

Do you have any ethical concerns with this paper?

Yes

Comments to the Author

This is a great paper. I commend the authors for doing such a deep and experimentally challenging job. Some minor points that appear in the comments on the manuscript require that attention of the authors. But basically it is almost ready for publication

Decision letter (RSPB-2020-0620.R0)

14-May-2020

Dear Dr LeKieffre

I am pleased to inform you that your manuscript RSPB-2020-0620 entitled "Ammonium is the preferred source of nitrogen for planktonic foraminifer and their dinoflagellate symbionts." has been accepted for publication in Proceedings B. Congratulations!!

The referee(s) have recommended publication, but also suggest some minor revisions to your manuscript. Therefore, I invite you to respond to the referee(s)' comments and revise your manuscript. Because the schedule for publication is very tight, it is a condition of publication that you submit the revised version of your manuscript within 7 days. If you do not think you will be able to meet this date please let us know.

- 1) A text file of the manuscript (doc, txt, rtf or tex), including the references, tables (including captions) and figure captions. Please remove any tracked changes from the text before submission. PDF files are not an accepted format for the "Main Document".
- 2) A separate electronic file of each figure (tiff, EPS or print-quality PDF preferred). The format should be produced directly from original creation package, or original software format. PowerPoint files are not accepted.

3) Electronic supplementary material: this should be contained in a separate file and where possible, all ESM should be combined into a single file. All supplementary materials accompanying an accepted article will be treated as in their final form. They will be published alongside the paper on the journal website and posted on the online figshare repository. Files on figshare will be made available approximately one week before the accompanying article so that the supplementary material can be attributed a unique DOI.

Sincerely,

Dr John Hutchinson, Editor

Associate Editor

Board Member: 1

Comments to Author:

You will see that the manuscript has been viewed by three reviewers which all see strong merit for this meticulous study.

The two latter reviewers have some suggestions for clarity and improvement, which should be considered.

Congratulations on this fine study.

Reviewer(s)' Comments to Author:

Referee: 1

Comments to the Author(s)

The target is clear, experiment was well designed, and the results are also clear and informative.

Referee: 2

Comments to the Author(s)

I only have a couple of minor editorial suggestions:

- 1) In the introduction, I would clarify (or rather, elaborate in a bit more details) the role of foraminifera as a component of primary production: while they do rely on heterotrophy for nitrogen, they are part of PP due to carbon fixation by symbionts
- 2) be consistent in presenting number formats (e.g. lines 261 and 270)
- 3) change to "highly ^{15}N -enriched..." (line 388)
- 4) I would like to see a sentence or two in the conclusions addressing the difference in $\delta^{15}\text{N}$ between dinoflagellate-bearing foraminifera and those with pelagophyte symbionts, which was brought up in the intro (Lines 67-69). In light of your findings, can you speculate what could drive the difference?
- 5) It is interesting that metabolic products from symbionts have higher isotopic values than the diet (lines 55-56). I think it might warrant a sentence or two of explanation.

Referee: 3

Comments to the Author(s)

This is a great paper. I commend the authors for doing such a deep and experimentally challenging job. Some minor points that appear in the comments on the manuscript require that attention of the authors. But basically it is almost ready for publication

Author's Response to Decision Letter for (RSPB-2020-0620.R0)

See Appendix A.

Decision letter (RSPB-2020-0620.R1)

27-May-2020

Dear Dr LeKieffre

I am pleased to inform you that your manuscript entitled "Ammonium is the preferred source of nitrogen for planktonic foraminifer and their dinoflagellate symbionts." has been accepted for publication in Proceedings B.

Open Access

Paper charges

Sincerely,

Appendix A

Dear Editor – we have greatly appreciated the constructive comments by the three reviewers and have modified our manuscript accordingly. Below we detail the changes made to the text in direct response to the reviewer's suggestions. After a new, more accurate, estimation of our manuscript length and because of the modifications we have made to our manuscript according to the reviewers' recommendations, our manuscript was exceeding the strict limit of 10 printed pages imposed by the Proceedings B. We first decided to move figure 2 in the supplementary file. We think that this figure presenting qualitative results of the carbon assimilation is not essential to the understanding of our manuscript, as the carbon assimilation by the symbiotic foraminifer species *Orbulina universa* is not the focus of the present study. Besides the quantitative data about carbon assimilation will still be presented in the main body of the manuscript (new figure 3, figure 4 in the previous version). However, this was not sufficient to reach the limit of 10 pages and we were recommended by the editor to suppress another 650 words from the manuscript. As the manuscript was accepted with minor revisions, we felt uncomfortable to make large modifications to the result or discussion sections that were not requested by the reviewers, nor initially by the editor. Thus, we have made the decision to suppress most of the text from the material and method section. Most of the methods used in this manuscript were previously described in previous papers that are cited in the text. Besides, all the text removed from the main body of the manuscript was placed in the supplementary, so the reader can still access it.

We feel that, with these modifications, the paper has been substantially improved and hope that the current version satisfies the requirements for publication in Proceedings B.

Sincerely Yours,

Charlotte LeKieffre

(On behalf of all co-authors)

Referee: 2

1) In the introduction, I would clarify (or rather, elaborate in a bit more details) the role of foraminifera as a component of primary production: while they do rely on heterotrophy for nitrogen, they are part of PP due to carbon fixation by symbionts

Reply: A couple of sentences were added in the introduction (lines 44 to 52 in the manuscript with tracked changes) to elaborate on the role of planktonic foraminifera in primary production.

2) be consistent in presenting number formats (e.g. lines 261 and 270)

Reply: The text was modified line 270 and every place it was necessary

3) change to "highly ^{15}N -enriched..." (line 388)

Reply: text was modified.

4) I would like to see a sentence or two in the conclusions addressing the difference in $\delta^{15}\text{N}$ between dinoflagellate-bearing foraminifera and those with pelagophyte symbionts, which was brought up in the intro (Lines 67-69). In light of your findings, can you speculate what could drive the difference?

Reply: this comment is related to the last comment of reviewer 3 and is discussed below.

5) It is interesting that metabolic products from symbionts have higher isotopic values than the diet (lines 55-56). I think it might warrant a sentence of two of explanation.

Reply: While this is a very interesting question based on modeling results from Uhle's paper (1997), this is not something we address in the paper as we are dealing with artificial enriched tracers. The readers can find more information if needed in the cited manuscript.

Referee: 3

- Line 81: Note deletions

Reply: text was deleted

- Line 204: vacuolated?

Reply: as "vacuolarized" is a term commonly found in literature, we prefer to keep it.

- Line 214: I still see more ^{13}C incorporation in Exp1 but may be only in the Figure shown.

Reply: It may seem on the graphs (new figure 3, previous figure 4) that some averaged data are higher for Exp. 1 than Exp. 2 but results from statistical analysis showed that the ^{13}C -enrichments were not significantly different, except for the lipid droplets and the dinoflagellates at $t = 1\text{h}$ as specified in the text.

- Line 222: I have the impression that more ^{13}C remains in the NH_4Cl (EXP 1) than in the NaNO_3 (Exp 2). Since the N species is the only difference this may be interesting. Perhaps the availability of ammonium enhances photosynthesis relative to nitrate.

Reply: We agree that comparing the C assimilation depending on the N source is interesting and surely requires further studies to confirm our observations. However, the experiments performed in the present study show no statistically significant differences between the two incubations for C assimilation (except for the first time point as mentioned in the previous comment and in the text). The putative effects of the available N-source on C assimilation are already briefly discussed lines 304 to 313. As the effects observed in our study were very minor and limited to the first time point, we do not feel that it is necessary to discuss it further.

- Lines 457 to 464 (Conclusion): This section should probably be part of the discussion and not in the conclusion. The way it is presented here there is not much information from the tracer experiments that can teach us on the natural fractionation. Obviously if a closed system is assumed then the fractionation will be small. The prey in nature is most probably copepods or other phytoplankton. These organisms are usually secondary producers which must have higher ^{15}N than the ammonium or nitrate of the primary producers. I guess this subject requires discussion and perhaps citation on the natural levels of N isotopes in planktonic forams (Ren's papers).

Reply: We have added information about what information our tracer experiments can bring to interpret natural fractionation recorded in paleo samples, in particular, to address the difference in $d^{15}\text{N}$ between

dinoflagellate-bearing foraminifera and those with pelagophyte symbionts (lines 474 to 476 in the manuscript with tracked changes). We believe that these modifications will satisfy Reviewer 2 and, in part, Reviewer 3. Reviewer 3 is asking for a discussion not only addressing species difference, but also why in natural environments, the species with dinoflagellate have $\delta^{15}\text{N}$ similar to nitrate. The second issue bears the implication that foraminifera must prey on food sources that isotopically represents the total exported N in the oligotrophic surface ocean. Discussions on this topic will be needed eventually, but we are a few steps behind and prefer not to make uncertain assumptions (some of the co-authors involved in the present study are currently working on new experiments and datasets that should address these specific questions in future publications).